# Study on the Performance of Steel Slag and Its Asphalt Mixture with Oxalic Acid and Water Erosion

**DOI:** 10.3390/ma15196642

**Published:** 2022-09-25

**Authors:** Xiaoming Huang, Feng Yan, Rongxin Guo, Huan He

**Affiliations:** 1Yunnan Key Laboratory of Disaster Reduction in Civil Engineering, Faculty of Civil Engineering and Mechanics, Kunming University of Science and Technology, Kunming 650500, China; 2Faculty of Environmental Science and Engineering, Kunming University of Science and Technology, Kunming 650500, China

**Keywords:** steel slag, oxalic acid, expansion, water erosion, asphalt mixture, road performance, influence mechanism

## Abstract

The reuse of steel slag, a large-scale solid waste from steel production, has good social and environmental benefits. The application of a steel slag asphalt mixture is mainly hindered by its volume expansion in water. The expansion of steel slag can be inhibited by oxalic acid. The expansion rate and adhesion of steel slag were investigated, and the immersion stability of steel slag and its asphalt mixture was evaluated by water erosion. By means of XRD, XRF, TG, SEM, etc., the influence mechanism of oxalic acid and water erosion on the properties of steel slag and its asphalt mixture was discussed. The results show that oxalic acid can not only inhibit the expansion of steel slag but also improve its crush resistance, with a reduction in the expansion rate of steel slag by 53%. Oxalic acid is able to leach alkaline metal elements, reducing its adhesion with asphalt. After 10 days of water erosion, the rutting stability and bending crack resistance of the treated steel slag mixture decreased by 37% and 43.2%, respectively. Calcium oxalate is generated on the surface of treated steel slag, which improves the surface compactness, effectively inhibits the expansion of steel slag caused by water erosion, and improves the performance of steel slag and its asphalt mixture. Water erosion can accelerate the hydration and shedding of calcium-containing substances on the surface of steel slag, reduce the adhesion of steel slag, and lead to degradation in the performance of steel slag and its asphalt mixture. Oxalic acid is able to effectively inhibit the expansion of steel slag, and the treated steel slag can be used as recycled aggregate in asphalt mixture, effectively solving the problems of road aggregate deficiency and environmental pollution caused by steel slag.

## 1. Introduction

Steel slag accounts for a large amount of the solid waste produced by steel making. With the development of China’s steel industry, the output of steel slag in China was as high as 100 million tons in 2021, and the total accumulated amount is as high as 1 billion tons at present. The multipurpose use rate of steel slag in China is less than 30%, of which less than 10% is used in building materials. Most steel slag is regarded as industrial waste, so it is filled or accumulated, resulting in ecological environmental pollution and other problems [1,2,3,4,5]. At the same time, China’s roads are in a stage of rapid development. By the end of 2021, the total mileage of expressways had exceeded 161,000 km. The problem of shortages of aggregate, as the main material from which expressways are constructed, has become more and more serious. Meanwhile, steel slag and natural aggregates have similar properties and can be mixed for use as aggregates. A large number of domestic laboratories’ research on steel slag asphalt mixtures and test sections of steel slag asphalt mixtures have demonstrated that the use of steel slag in road materials is the primary means of eliminating them [6,7,8,9].

The steel slag accumulated and produced in China has the property of volume expansion. The main factor affecting the expansion of steel slag is free calcium oxide (f-CaO), which undergoes a slow hydration reaction when coming into contact with water. The process of generating calcium hydroxide (Ca(OH)_2_) leads to the expansion and cracking of steel slag. There is a close relationship between f-CaO and the expansion of steel slag, and f-CaO on the surface of steel slag is more likely to cause expansion and cracking than that inside [10,11]. Road researchers have focused on the treatment of steel slag, and the reuse of steel slag after treatment has become an industry trend [12,13,14]. The main effective treatment methods include carbonization, acid treatment, surface coating, etc. The carbonization of steel slag effectively reduces the volume expansion of steel slag, while at the same time improving its hydration activity [15,16]. The acid treatment effectively reduces the expansion of steel slag. Commonly used acids include glacial acetic acid, oxalic acid, phosphoric acid, formic acid, etc. The main treatment methods include chemical dry modification, chelating treatment, etc., which can reduce the expansion of steel slag, and improve the activity of steel slag powder [17,18,19]. Treated steel slag is mainly applied in asphalt mixture. Coating resin, cement slurry, waterproofing agent, and other materials, when applied on the surface of steel slag, can hinder the contact between steel slag and water, thus inhibiting the expansion of the steel slag [20,21]. These methods are not able to effectively eliminate f-CaO in steel slag, and only inhibit the hydration of f-CaO. During the subsequent application process, the coating layer may fall off, resulting in the subsequent expansion of the steel slag. The long-term application of treated steel slag has not been verified and may lead to potential safety hazards. The immersion stability of steel slag asphalt mixture under the action of water is worth considering.

To solve the problem of the volume expansion of steel slag caused by water, and to evaluate the immersion stability of treated steel slag and its asphalt mixture, an oxalic acid solution was used to treat steel slag aggregate. This paper analyzes the mechanism of influence of oxalic acid and water erosion in the performance of steel slag and its asphalt mixture, evaluates the applicability of steel slag and its asphalt mixture following treatment with oxalic acid, and provides engineering suggestions and theoretical guidance for the practical application of steel slag in asphalt mixture.

## 2. Materials and Methods

### 2.1. Materials

BOF steel slag aggregate, limestone aggregate, and SBS(I-D) polymer modified asphalt were used as raw materials. The raw material used for treatment was a self-prepared oxalic acid solution; oxalic acid was produced by Tianjin Chemical Three Plant Co., Ltd. with the chemical formula C_2_H_2_O_4_·2H_2_O and a reagent content of not less than 99.6%, as a colorless transparent crystal or powder. The chemical composition of the selected steel slag, and its basic physical properties, are shown in Table 1 and Table 2. Moreover, some properties of different steel slags and natural aggregates are compared. The main components of steel slag are calcium oxide, iron oxide, and silica, accounting for more than 10%, while mechanical properties such as Los Angeles abrasion have quite different values, which is mainly a result of the different processes employed in different steel mills [22,23]: SSD is pristine steel slag, which was provided by Shandong Iron and Steel Co., Ltd., while EOF steel slag is energy-optimized furnace slag. The basic physical properties of limestone and the experimental results of SBS-modified asphalt are shown in Table 3 and Table 4. The physical indexes of fine limestone aggregate and asphalt meet the specification requirements. However, the density of fine limestone aggregate is generally lower than that of steel slag—below 3 g/cm^3^. This is mainly because there is a large amount of iron in steel slag, which is one of the main reasons for the high density of steel slag [2].

### 2.2. Treatment Method

Steel slag is mainly used as coarse and fine aggregate in asphalt mixtures; therefore, 0–16 mm steel slag was used for the treatment. Different concentrations of oxalic acid solutions were prepared, and different treatment times were adopted for the preliminary steel slag treatment. The best treatment concentration was determined on the basis of the water absorption rate and the f-CaO content of treated steel slag, and the best treatment time was determined on the basis of the change in pH value. In the treatment process, the steel slag to be treated was put into the prepared 0.4 mol/L oxalic acid solution at 25 °C according to a solid–liquid ratio of 1:5. To avoid the CO_2_ and water in the air influencing the treatment process, the treatment process was sealed after stirring for 3 min, and the treatment time was 84 h. Treated steel slag can be used after being dried at 105 °C. The specific treatment process is as follows (Figure 1).

### 2.3. Experimental Method

According to the Test Method of Aggregate of Highway Engineering (JTG E42-2005), the basic physical properties of steel slag and limestone were tested. The basic index of SBS-modified asphalt, the road performance index of the steel slag asphalt mixture, and the mix design of steel slag asphalt mixture were adopted with reference to the Standard Test Methods of Bitumen and Bituminous Mixtures for Highway Engineering (JTG E20-2011). There are three index samples in each group tested by steel slag and its asphalt mixture, and the indexes of the samples need to meet the specifications. The experimental results are the average of the results of the three samples. The water stability of the mixture is tested by a Universal testing machine (UTM-30), including the Marshall stability test and freeze-thaw splitting test. The sample in the Marshall stability test needs to be kept in a water bath at a constant temperature of 60 °C for 30–40 min and 48 h, respectively. After the heat preservation is completed, it is put into a fixture for testing, and the loading rate is 50 mm/min. For the samples in the freeze–thaw test, the molded samples are divided into two groups. One group is stored at room temperature, and the other group is kept in a vacuum for 16 h. After taking out, the samples are kept in a water bath at 60 °C for 24 h. Finally, the samples of both groups are kept in a water bath at 25 °C for 2 h, and then taken out for testing. The loading rate is 50 mm/min. The rutting stability of the mixture is tested by a rutting tester (SYD-0719B). The formed sample is kept in a 60 °C incubator for 5 h, and then put into a test bench for testing. The test will be stopped when the test time is 1 h or the maximum deformation is 25 mm.

The mineral composition and phase composition of steel slag have a great influence on the properties of steel slag. To ensure the average of the selected steel slag, selected the original steel slag by the quartering method and ground it to below 0.075 mm. The material was tested by X-ray diffraction (XRD), the diffraction pattern was analyzed, the mineral composition of the steel slag was obtained, the equipment used in this experiment is Ultima IV, with the parameters of X-ray Cu Kα Radiation Diffraction Meter, wavelength of 0.15406 nm, voltage of 40 KV, current of 40 mA, testing speed of 5°/min, and testing angle of 10–90. X-ray fluorescence (XRF) is used to test the elements and substance composition in steel slag, and the equipment model is PANalytical Axios. Moreover, the microstructure of the steel slag is observed using a scanning electron microscope (SEM), its equipment model is Hitachi Regulus8100. Ethylene glycol differential thermal analysis—EDTA-TG—is an effective current method that is accurate and simple for measuring the content of f-CaO in steel slag. After grinding the steel slag selected by the quartering method to below 0.075 mm, the free total calcium in the steel slag is determined by titration with ethylene glycol, and the free total calcium content obtained using this method includes the content of calcium hydroxide. The content of calcium hydroxide in steel slag is determined by TG, and its equipment model is TA. The content of f-CaO is obtained by subtracting the content of calcium hydroxide from the free total calcium [24]. The 10-day soaking expansion rate of steel slag was defined with reference to the Test Method for Stability of Steel Slag (GB/T 24175-2009). The specific process is as follows: the steel slag mixture is mixed with the optimal water content, compacted by a standard heavy-duty compactor, and then put into the volume expansion rate mold, as shown in Figure 2. The initial dial gauge is read, and the mold is soaked in a 90 °C water bath for 6 h to completely immerse the sample. Before heating, the daily dial indicator reading of the sample height is recorded for 10 days. According to Formula (1) [25],
(1)$$\gamma =\frac{d_{10}-d_{0}}{120}\text{×}100$$
where γ is the water immersion expansion rate, %; 120 is the original height of the test piece, mm; *d*_10_ is the final reading of the dial indicator, mm; *d*_0_ is the initial reading of the dial indicator, mm.

The application process of steel slag and its asphalt mixture is greatly affected by water, which also directly causes steel slag expansion. To intuitively evaluate the changes in the performance of steel slag and its asphalt mixture under the action of water, the water erosion method is adopted [10,11]. Water erosion process: the steel slag aggregate and the formed mixture specimen are placed in a normal-temperature water bath box, and the steel slag aggregate and the asphalt mixture need to be completely submerged by the water. After soaking for a long setting time, they are taken out for performance testing. The adhesion test [26] was used to evaluate the adhesion between aggregate and asphalt. Two representative groups of limestone were taken, combined with untreated steel slag and treated steel slag, and put in asphalt for 45 s. Then, they were cooled in the air for 15 min, and placed in boiling water for 3 min and 30 min; their adhesion was evaluated using Formula (2):(2)$$S=\frac{m_{1}-m_{2}-m_{3}}{m_{1}-m_{2}}$$
where *S* is the anti-stripping percentage of asphalt, *m*_1_ is the original mass of asphalt aggregate, *m*_2_ is the original mass of aggregate, and *m*_3_ is the loose loss mass of the aggregate surface.

### 2.4. Mix Proportion Design of Steel Slag Mixture

At present, there are many specifications of asphalt mixture, and AC asphalt mixture is widely used in pavement. AC-13c is the main type in most experiments [12]. AC-13c asphalt mixture is a continuous gradation of fine particles, with a nominal maximum particle size of 13 mm. Therefore, this paper uses the AC-13c Marshall preparation method to prepare steel slag asphalt mixture [27].

As oxalic acid cannot directly affect the particle size of steel slag, the gradation curves of steel slag before and after treatment show little difference. However, in order to better standardize the experimental results, designed two gradations under the condition of meeting the lower limit of gradation. The specific gradation curves are shown in Figure 3. In addition, the percentage passing through the graded mesh quality is shown in Table 5. The gross bulk density, stability, flow value, and volume index of the two kinds of asphalt mixtures prepared with five asphalt–aggregates ratios (4.3%, 4.7%, 5.2%, 5.7%, and 6.2%) were tested. In addition, the best asphalt–aggregate ratios for steel slag asphalt mixtures before and after treatment were determined to be 5.2% (mass fraction) and 4.8% (mass fraction), respectively. The indexes of the Marshall specimen of the steel slag asphalt mixture under the best asphalt–aggregate ratio are shown in Table 6.

This section mainly discusses the gradation design and the best asphalt–aggregate ratio of steel slag asphalt mixture before and after treatment. The gradation curve of steel slag asphalt mixture before and after treatment has little difference. It is worth noting that the best asphalt–aggregate ratio of steel slag asphalt mixture after treatment has decreased by 0.4%. All calculated data are the average value from three tests.

## 3. Results and Discussion

### 3.1. Basic Properties of Steel Slag Aggregate

#### 3.1.1. Expansion of Steel Slag

The main factors causing the volume expansion of steel slag are f-CaO, free magnesium oxide (f-MgO), and RO phase. f-CaO, the f-CaO caused volume expansion corresponding to 91.7% [28,29]. Because the determination methods of f-MgO are not uniform, the current determination methods are not able to achieve rapid and accurate results, and there is no specific method for the determination of RO phase, the f-CaO content of steel slag aggregate and the soaking expansion rate are the main indexes for judging the volume stability of steel slag [2]. Table 7 and Figure 4 show the changes in f-CaO content, expansion rate, and Ca(OH)_2_ content of steel slag before and after treatment, respectively. After treatment with oxalic acid, the content of f-CaO decreased from 4.551% to 2.225%, and the expansion rate decreased from 3.59% to 1.69%. The mass loss of Ca(OH)_2_ at 400~550 °C was calculated by TG analysis. The mass loss of water in steel slag at 400–550 °C can be expressed as the loss of calcium hydroxide [28]. After the steel slag is treated with oxalic acid, the content of calcium hydroxide is increased by 127%, which indicates that oxalic acid is able not only to consume f-CaO, but also promote its hydration. The soaking expansion rate of steel slag can intuitively indicate its volume stability. Formula (3) is used to fit the 10-day soaking expansion rate of steel slag [30]. In the early stage of steel slag soaking, the expansion rate increased greatly, because, under the conditions of the 90 °C water bath, f-CaO hydrates to form Ca(OH)_2_, causing volume expansion. With the passage of time, the hydration reaction weakened, the content of f-CaO decreased, and the growth trend of expansion rate decreased [31]. In the process of treating steel slag with oxalic acid, it reacts with some basic oxide f-CaO, effectively reducing the content of f-CaO in the steel slag, reducing the expansion rate and expansion speed of the steel slag, and effectively reducing the expansion of the steel slag.
(3)$$Y=a(1-e^{(-\mathrm{bx})})$$
where Y is the expansion rate of steel slag, %; a and b are constants; e = 2.71828183…; and x is time, day.

#### 3.1.2. Basicity and Adhesion

The adhesive force between aggregate and asphalt is one of the main factors determining the performance of the mixture. Because steel slag is a basic aggregate, there are a lot of metal cations such as Ca^2+^, Mg^2+^, Al^3+^, Fe^3+^, and Mn^2+^ on the surface of steel slag. In the process of its combination with asphalt, it can react with some substances in asphalt, such as asphaltene groups, to generate bituminous acid salt, and form an adsorption layer on the surface of steel slag, which has high chemical strength. Therefore, steel slag and asphalt have good adhesive force [13,32]. The asphalt mixture prepared using it has good performance. It is worth noting that in the process of treating steel slag with oxalic acid, the basicity of the steel slag may be decreased due to acid–base reaction, and the adhesive force may be reduced, such that there is a reduction in the performance of the mixture. Therefore, by comparing the basicity of steel slag and limestone before and after treatment, and the adhesion with asphalt, the adhesion between the steel slag and asphalt after oxalic acid treatment was tested. Table 8 presents a comparison of the chemical composition and basicity between the steel slag and the limestone before and after treatment. After treatment, the content of calcium oxide (CaO) and iron oxide (Fe_2_O_3_) in the steel slag decreases, while the content of silicon dioxide (SiO_2_) changes little, and the content of other metal oxides decreases slightly. After treatment, the basicity of steel slag decreases slightly, but it is still higher than that of limestone. Figure 5 shows the results of the adhesion test. Compared with untreated steel slag, there was a decrease in the adhesion of treated steel slag at 3 min and 30 min, and the asphalt peeling rate increased by 33.6% and 8%, respectively. However, the peeling rate of treated steel slag at 3 min and 30 min was lower than that of limestone, and its adhesion was still better than that of the limestone aggregate. It has been proved that oxalic acid is able to leach calcium, iron, and other metal ions from steel slag [33,34], and can be combined with it to form oxalic acid salt. The surface basicity of steel slag is reduced, the reaction with acid groups in asphalt is weakened, and the adhesive force is reduced, resulting in a decrease in its adhesion.

#### 3.1.3. Crush Resistance

At present, most countries use abrasion to express the anti-crushing ability of aggregate, but the importance of crush value in the asphalt mixture cannot be ignored. Therefore, in this section, the abrasion value and the crush value are used to jointly characterize the anti-crushing ability of the aggregate. Figure 6 shows the changes in abrasion value and crush value of steel slag aggregate with different water erosion times. After treatment with oxalic acid, the crush resistance of steel slag is improved, and the abrasion value and crush value are reduced by 27.7% and 7.9%, respectively. With increasing water erosion time, the crush resistance of steel slag before and after treatment showed a decreasing trend, but the abrasion and crush value of treated steel slag gradually became stable. After 10 days of water erosion, the abrasion and crush values of untreated steel slag increased by 2.4% and 3.1%, respectively, while those of treated steel slag increased by 2.1% and 3.3%, respectively. The overall decrease was small and the impact on the crush resistance of steel slag was low. Figure 7 shows the pH change in steel slag eroded by water. Under the long-term erosion of steel slag aggregate and water, silicate minerals and CaO in the steel slag aggregate undergo a hydration reaction with water, resulting in hydrated calcium silicate (C-S-H) and Ca(OH)_2_. During the erosion process, the pH value changes obviously, and CO_2_ in the air reacts with Ca(OH)_2_ to generate calcium carbonate (CaCO_3_), resulting in a white coating on the surface of the aqueous solution [35], as shown in Figure 8. Additionally, with increasing water erosion time, the amount of white coating gradually increases, and the pH value gradually increases. In the process of hydration to form C-S-H and Ca(OH)_2_, the volume of steel slag aggregate expands, micro-cracks appear on the surface and pores of steel slag, and water immerses into the cracks, providing further hydration. However, due to the slow hydration process, only the surface can be hydrated, resulting in a slight decrease in the strength of the steel slag aggregate [36]. Under conditions of water erosion, the pH value of steel slag treated with oxalic acid changes slightly, the surface of the aqueous solution is clear, and its pH value exhibits a stable trend. In the process of treating steel slag with oxalic acid, most silicate minerals and f-CaO on the surface of steel slag are acidified to generate oxalic acid salt, reducing the hydration activity of the steel slag surface. With increasing water erosion time, the hydration between steel slag and water decreases, and the steel slag and water solution remain in a relatively stable state.

### 3.2. Water Stability of Steel Slag Asphalt Mixture

#### 3.2.1. Marshall Stability

Table 9 shows the change in the stability of the steel slag asphalt mixture before and after the treatment with oxalic acid under different water erosion times, and Figure 9 shows the change in residual stability. Generally speaking, the residual stability of steel slag treated with oxalic acid is slightly better than that of untreated steel slag under the action of water erosion at different times. It is worth noting that the residual stability of steel slag before and after the treatment is mostly higher than 100%, which indicates that the strength of the mixture is improved after heat preservation for 48 h at 60 °C. The residual stability of the untreated steel slag increased gradually while undergoing erosion for durations of 0–4 days, but decreased for durations of 4–10 days of erosion, with an obvious decrease of 15.7%. After treatment, the residual stability of steel slag can be maintained at about 105%, being almost unaffected by water erosion. Steel slag with f-CaO content higher than 3% undergoes expansion, and the f-CaO content of the steel slag selected in this study was as high as 4.852%. Long-term immersion causes water to infiltrate into the asphalt, coming into contact with the steel slag. During the research process, it is necessary to keep the temperature in the water bath at 60 °C for 0.5 h or 48 h, causing the asphalt film on the surface of the steel slag to peel off, and the f-CaO on the surface is more likely to react with the water in order to generate Ca(OH)_2_, resulting in the expansion of the steel slag [36], thereby filling the internal pores of the mixture, and improving the compactness and the resistance to deformation of the mixture. However, after being eroded by water for more than 4 days, most of the f-CaO on the surface of untreated steel slag continues to hydrate, and the hydration activity of the steel slag aggregate improves. After a 60 °C water bath, the hydration reaction is intensified, and the internal expansion of the steel slag aggregate is too great [31], which leads to the slow destruction of its structure, thus reducing its residual stability. Compared with untreated steel slag, oxalic acid is able to eliminate most silicate minerals and f-CaO on the surface of the steel slag and inhibit the hydration reaction, resulting in an insignificant change in the overall residual stability.

#### 3.2.2. Freeze–Thaw Splitting

Table 10 and Figure 10, respectively, show the changes in the freeze–thaw splitting strength and strength ratio of steel slag asphalt mixture before and after the treatment under different water erosion conditions. Under early water erosion conditions, the unfrozen and freeze–thaw splitting strength of the treated steel slag is higher than that of untreated steel slag. However, with increasing water erosion time, the splitting strength of treated steel slag gradually becomes lower than that of untreated steel slag. By observing the freeze–thaw splitting strength ratio of steel slag asphalt mixture before and after treatment, it is found that the freeze–thaw splitting strength ratio of treated steel slag is lower than that of untreated steel slag, and after 4 days of water erosion, there is an increase in the decreasing trend of the freeze–thaw splitting strength ratio. Generally speaking, the freeze–thaw splitting strength ratio of steel slag asphalt mixture before and after treatment meets 75% of the specification requirements. With increasing water erosion time, there was a decrease in the freeze–thaw splitting strength and splitting strength ratio of the steel slag asphalt mixture. Compared to the Marshall stability test, the freeze–thaw splitting test conditions are more stringent, making them better able to reflect the water damage resistance of the mixture. With increasing water erosion time, the emulsification between water and asphalt intensifies, and the asphalt layer peels off, leading to the overall structure of the mixture becoming loose. However, the vacuumizing process is able to fill most of the empty pores of the mixture, and through vacuumizing and freezing, the expansion effect is weakened, the asphalt shrinks and becomes brittle, the brittleness of the mixture is increased, and the adhesive force between the asphalt and the aggregate decreases [37]. However, the treated steel slag surface precipitates and transforms, resulting in calcium oxalate, which improves the strength of the steel slag [38,39], and under the condition of water erosion, the expansion effect of hydration reaction of steel slag in the water bath is inhibited, so the ratio of freeze–thaw cracking strength and splitting strength of treated steel slag decreases rapidly.

### 3.3. Rutting Stability and Bending Crack Resistance of Steel Slag Asphalt Mixture

Under the same conditions with respect to gradation, aggregate, and asphalt type, the rutting stability of the mixture is mainly related to its strength and compactness. The higher the strength and compactness of the mixture, the higher its dynamic stability and the better its rutting stability [18,40]. Figure 11 shows the changes in dynamic stability and maximum bending strain of the steel slag mixture before and after treatment for different water erosion times. The dynamic stability of the steel slag before and after treatment decreased by 28.4% and 37%, respectively, after 10 days of water erosion, and the maximum bending strain decreased by 26.7% and 43.3%, respectively. It can be clearly seen that the rutting stability of treated steel slag is effectively improved, and the dynamic stability of the steel slag gradually decreases with increasing water erosion time. In the process of water erosion of the mixture, water emulsifies the asphalt, and the adhesion gradually decreases. Oxalic acid reduces the volume expansion of steel slag, but it can also reduce adhesion. Although most of the f-CaO on the surface of the steel slag is eliminated, under the condition of long-term immersion, the asphalt film keeps falling, the compactness decreases, and the bonding degree decreases [41]; however, the strength of the mixture will still decrease. This is the main reason for which the rutting stability of the steel slag mixture decreases with increasing water erosion time. At present, the influencing factors of the bending crack resistance of the mixture are not clear, and most researchers think that the bending stability of the mixture is related to its adhesion [42,43]. With increasing water erosion time, the bending crack resistance of the mixture gradually decreases. In the process of erosion for 0–1 day, the maximum failure strain of treated steel slag is higher than that of untreated steel slag. After 4 days of water erosion, it can be observed that the maximum failure strain of the treated steel slag is lower than that of the untreated steel slag, which corresponds to the freeze–thaw splitting strength reported in the previous section. The results show that the performance of the steel slag asphalt mixture treated with oxalic acid will be greatly reduced after a short duration of water erosion and in a low–temperature experiment. The authors believe that the decrease in the adhesion between asphalt and steel slag is mainly due to water erosion. The asphalt shrinks and becomes brittle, aggregate pores increase, and finally, its bending crack resistance decreases [40]. After the treatment, the metal ions on the surface of steel slag combined with asphalt decrease, resulting in its adhesion with asphalt decreasing rapidly with water erosion.

### 3.4. Mechanism of Action of Oxalic Acid

Figure 12 presents a comparison diagram of the mineral composition of steel slag before and after treatment with oxalic acid. Steel slag is a hydraulic mineral with complex mineral composition, mainly including tricalcium silicate (Ca_3_SiO_5_), dicalcium silicate (Ca_2_SiO_4_), calcium ferrite (Ca_2_Fe_2_O_5_), and RO. After the steel slag is treated with oxalic acid, the diffraction peak of calcium oxalate (CaC_2_O_4_) appears. Some of the Ca_2_SiO_4_ and Ca_3_SiO_5_ diffraction peaks disappear and decrease. The diffraction peak of CaO disappears and changes to Ca(OH)_2_. The presence of oxalic acid breaks the acid–base balance of the original system of steel slag, and some silicate minerals on the surface of the steel slag, such as Ca_2_SiO_4_ and Ca_3_SiO_5_, react with oxalic acid to form the CaC_2_O_4_ complex. During the treatment of steel slag, the hydration speed of f–CaO accelerates, and Ca(OH)_2_ is generated. Under the closed treatment condition, CO_2_ in the air fails to participate in the reaction, so there is Ca(OH)_2_ after the treatment, which is consistent with the quality loss analysis results Ca(OH)_2_ described above. Figure 13 presents a comparison of the SEM results of the surface of the steel slag. The surface of the untreated steel slag is complex and diverse, consisting of mostly spherical weak particles (CaCO_3_), accompanied by a large number of loose network structures [44]. A large number of square blocks appear on the surface of the treated steel slag, which are judged to be Ca_2_C_2_O_4_ crystals [45,46], accompanied by a small amount of flake Ca(OH)_2_ [47]. Steel slag has a porous structure. During long-term treatment with oxalic acid, oxalic acid can acidify the surface of most steel slag, and penetrates into the pores of steel slag, forming CaC_2_O_4_ complexes on the surface and in the pores. The weak particles and reticular structure on the surface of the untreated steel slag are replaced and filled by the generated CaC_2_O_4_, resulting in precipitation transformation. The surface compactness of the steel slag increases and the strength of the steel slag increases slightly [48]. Oxalic acid can also promote the hydration reaction of f–CaO and react with some f–CaO in steel slag, which effectively reduces the expansion of the steel slag. The reaction principle is shown in Equations (4)–(6).

During the treatment of steel slag with oxalic acid, a large number of calcium ions dissolve out and react with oxalic acid to generate Ca_2_C_2_O_4_:(4)$${\mathrm{Ca}}^{2+}{+H}_{2}C_{2}O_{4}{=\mathrm{CaC}}_{2}O_{4}{+2H}^{+}$$

Oxalic acid can also promote the hydration of f-CaO:(5)$$f-\mathrm{CaO}+H_{2}O=\mathrm{Ca}{\left(\mathrm{OH}\right)}_{2}$$
(6)$$\mathrm{Ca}{\left(\mathrm{OH}\right)}_{2}+H_{2}C_{2}O_{4}={\mathrm{CaC}}_{2}O_{4}+{2H}_{2}O$$

In Ding et al.’s research [19], oxalic acid was used to treat steel slag fine aggregate, and the treated steel slag was used as aggregate in concrete. In the process of oxalic acid treatment, calcium oxalate generated by the reaction between oxalic acid and f-CaO and calcite in cement precipitated on the concrete surface, forming a dense protective layer, which improved the compactness of the concrete and reduced the expansion of the concrete samples, reducing its linear expansion from 4.7% to 1.7%. The compressive and flexural strength of the final concrete samples were improved. In Huo et al.’s research [17,18], they used formic acid and glacial acetic acid to treat steel slag powder by dry modification. Because formic acid and glacial acetic acid can react with mineral components on the surface of steel slag to generate calcium formate and calcium acetate, the surface of steel slag will be etched during the reaction process, resulting in improvements with respect to the specific surface area and roughness of the treated steel slag powder, an increase in the number of pores on the surface of the steel slag powder, and easier access of water into the steel slag powder. The early hydration speed of the formed cement paste is faster, resulting in more hydration products. However, part of the f-CaO in the steel slag can react with formic acid and glacial acetic acid, consuming part of the f-CaO. Due to the compactness of the hydration products, the volume stability of the cement paste is finally improved. The final result is that formic acid and glacial acetic acid are able to improve the hydration activity of steel slag powder so as to improve the volume stability. In this study, oxalic acid was used to treat steel slag, and the treatment principle was similar to that of Yung, Huo, and others; however, we obtained a more mature conclusion. Firstly, after treatment with oxalic acid, a large number of square calcium oxalate crystals are formed on the surface and pores of the steel slag, and the compactness of the steel slag surface is improved, providing a certain strength, which contributes to improving the road performance of the steel slag asphalt mixture. However, acid can accelerate the hydration of most substances in steel slag and consume part of the f-CaO in the steel slag. After acid treatment, the hydration activity of steel slag is greatly reduced, and the hydration degree of f-CaO in steel slag is reduced when it meets water again, finally leading to a significant decrease in the expansion rate of the steel slag.

### 3.5. Mechanism of Influence of Water Erosion on the Phase Composition of Steel Slag

Figure 14 shows the changes in the mineral composition of the steel slag before and after treatment after water erosion for 0 days, 5 days and 10 days, respectively. The intensity of most mineral diffraction peaks of steel slag decreases, while some peaks, such as Ca_2_Fe_2_O_5_, Ca_2_SiO4, and Ca_2_C_2_O_4_, decrease obviously, and are almost invisible in the XRD diffraction patterns. Most minerals on the surface of the steel slag can hydrate to form hydrated calcium silicate (C-S-H) during the process of water erosion, especially Ca_2_SiO_4_ and Ca_3_SiO_4_, which are invisible in the XRD diffraction patterns because of their amorphous forms [49,50,51]. Table 11 shows the changes in the chemical elements in the steel slag before and after the treatment with water erosion. Under the condition of water erosion, most of the elements in steel slag are decreased, as, for example in the cases of the Ca, Fe, Si, and other elements. Figure 15 shows the changes in the main chemical elements. The chemical elements in the steel slag after the treatment exhibit no obvious changes with water erosion, and the overall chemical elements tend to be stable compared with untreated steel slag, while the calcium element content decreases slightly, which is consistent with the analysis of mineral composition above. Figure 16 shows the change in the micro-morphology of steel slag before and after treatment with water erosion at different times. Water erosion can accelerate the hydration of calcium silicate and f-CaO on the surface of steel slag, resulting in floc C-S-H and flake Ca(OH)_2_ [52,53,54]. With increasing water erosion time, flake Ca(OH)_2_ and floc C-S-H on the surface of untreated steel slag become more and more obvious, and at the same time, a few microcracks appear. However, the surface of the treated steel slag is mostly massive Ca_2_C_2_O_4_, and water erosion can accelerate the shedding degree of Ca_2_C_2_O_4_. Moreover, floc C-S-H was not found on the surface of steel slag, which was due to the existence of a large amount of Ca_2_C_2_O_4_ blocking the C-S-H. With increasing shedding degree of Ca_2_C_2_O_4_, a small amount of fashioned Ca(OH)_2_ begins to appear on the surface of the steel slag, which indicates that the f-CaO on the surface of the treated steel slag has been basically eliminated, which is in agreement with the previous experimental results.

Most of the Ca_2_C_2_O_4_ present on the surface of the treated steel slag is free in aqueous solution due to the action of water erosion. With increasing water erosion time, the degree of Ca_2_C_2_O_4_ that is free and falling off increases, resulting in a slight decrease in the strength of the steel slag. At the same time, the shedding of the asphalt layer and the lack of basic metal content on the surface of steel slag have a great influence on adhesion, which leads directly to the obvious downward trend of the adhesion in the mixture with the action of water erosion, and the downward trend of the performance of the mixture [55,56].

## 4. Conclusions

In this study, a self-prepared oxalic acid solution was used to treat steel slag with the aim of inhibiting its expansion, and the steel slag and its asphalt mixture were subjected to water erosion at different times before and after treatment. Then, the basic properties of the steel slag and its asphalt mixture were tested by means of comparison. The mechanisms of influence of oxalic acid and water erosion on the mixed properties of steel slag and its asphalt mixture were discussed. The potential of steel slag treated with oxalic acid in an asphalt mixture was evaluated. The following conclusions were drawn:(1)Oxalic acid is able to promote the hydration reaction of f-CaO on the surface of steel slag, eliminate some f-CaO in steel slag, and inhibit the volume expansion of steel slag. However, due to the leaching of alkaline ions, the adhesion decreases. After 7 days of water erosion, the crush resistance of treated steel slag tends to be stable. Water erosion has little effect on the crush resistance of steel slag.(2)After oxalic acid treatment, the performance of the steel slag asphalt mixture is improved, but in the process of water erosion, its performance decreases obviously, especially with respect to rutting stability and bending crack resistance. The performance of the steel slag asphalt mixture after treatment is greatly reduced after being subjected to low temperature, and its low-temperature performance needs further consideration.(3)Water erosion is able to accelerate the hydration of calcium silicate and f-CaO on the surface of steel slag, leading to the slight expansion of the steel slag surface, decreased adhesion, and a decline in the performance of the steel slag asphalt mixture. Oxalic acid can react with calcium silicate and f-CaO on the surface of steel slag to produce Ca_2_C_2_O_4_. It can promote the hydration of f-CaO, filling the surface and most pores of the steel slag, effectively inhibiting the expansion effect of f-CaO hydration on the surface of steel slag after water erosion, and improving the strength of the steel slag.(4)The surface of steel slag treated with oxalic acid mostly consists of the Ca_2_C_2_O_4_ complex. With increasing water erosion time, the shedding degree of Ca_2_C_2_O_4_ increases. Most basic metal elements on the surface of steel slag are leached out, and the adhesion between steel slag and asphalt decreases more obviously, which finally leads to a decline in the performance of the asphalt mixture. After a long duration of water erosion, the properties of steel slag treated with oxalic acid and its asphalt mixture are similar to those of untreated steel slag, indicating that it is feasible to apply steel slag treated with oxalic acid in asphalt mixtures.(5)This work solves the problem that steel slag is difficult to apply due to volume expansion, and the treated steel slag can be widely used in asphalt mixtures. This provides a solution to the excessive accumulation and output of steel slag at present, in China, and solves the environmental problems caused by steel slag and the lack of natural aggregates in China. It is worth noting that whether the performance of steel slag asphalt mixture changes under the long-term action of water erosion and how the performance changes under actual application conditions are the factors that determine whether the steel slag asphalt mixture can ultimately be applied. In future research, it is worth our continued consideration.

## Figures and Tables

**Figure 1 materials-15-06642-f001:**
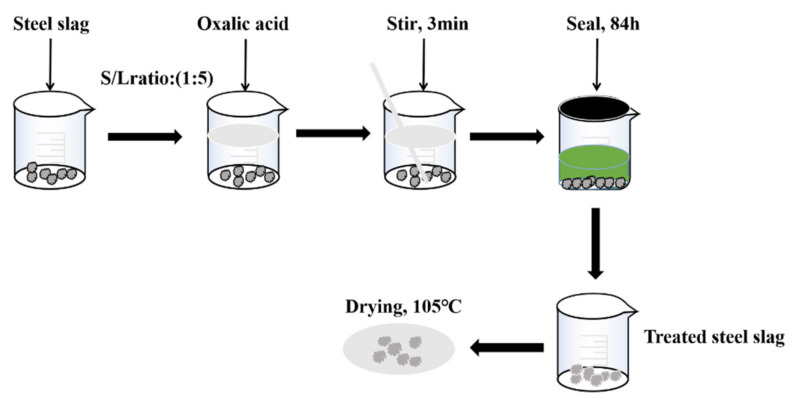
Treatment process of steel slag.

**Figure 2 materials-15-06642-f002:**
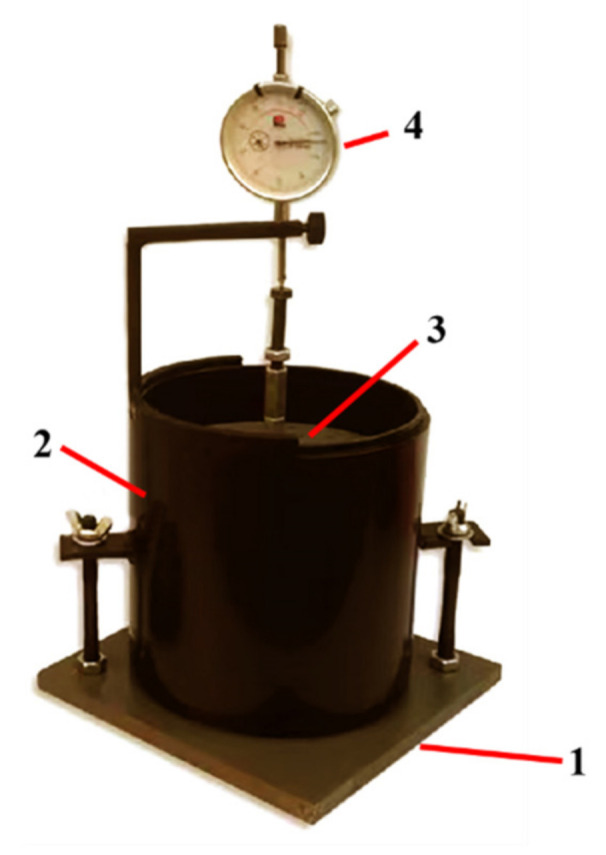
Steel slag volume expansion test setup. 1—Lower perforated plate; 2—steel slag; 3—surcharge board; 4—dial gauge.

**Figure 3 materials-15-06642-f003:**
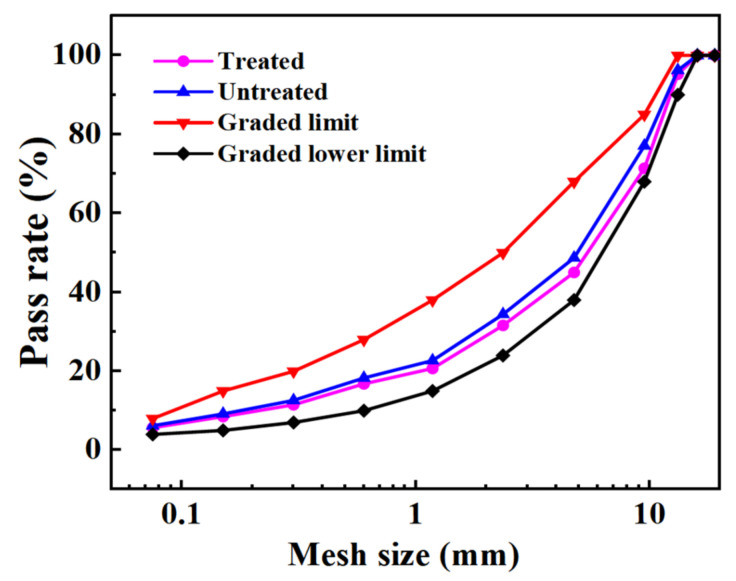
Asphalt mixture grading curve.

**Figure 4 materials-15-06642-f004:**
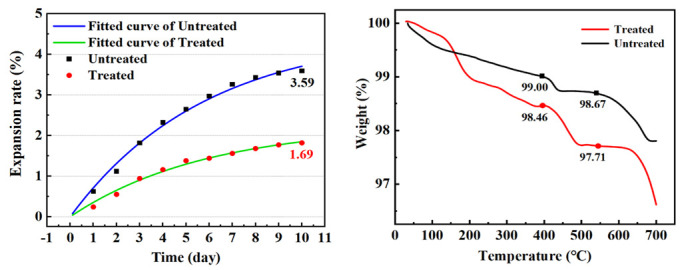
Comparison of steel slag expansion rate and Ca(OH)_2_ mass loss.

**Figure 5 materials-15-06642-f005:**
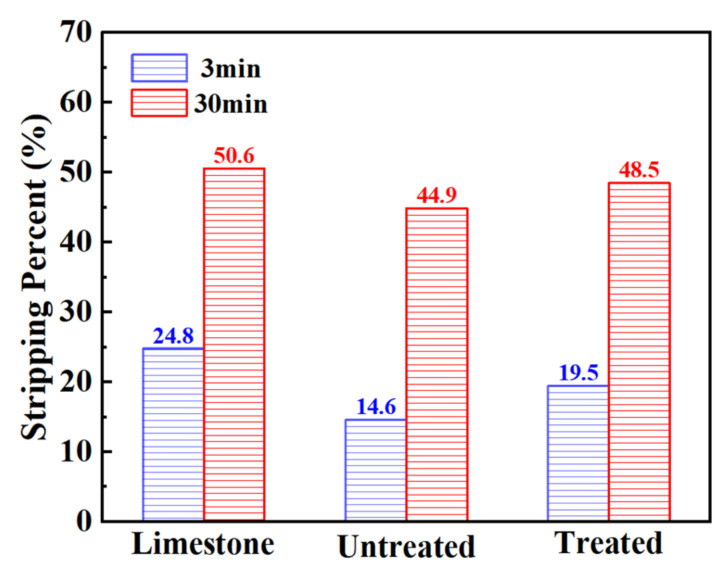
Boiling water stripping rate of aggregate.

**Figure 6 materials-15-06642-f006:**
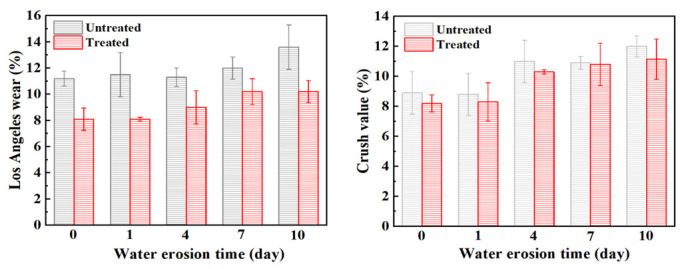
The Los Angeles wear and crush value of steel slag after different water erosion times.

**Figure 7 materials-15-06642-f007:**
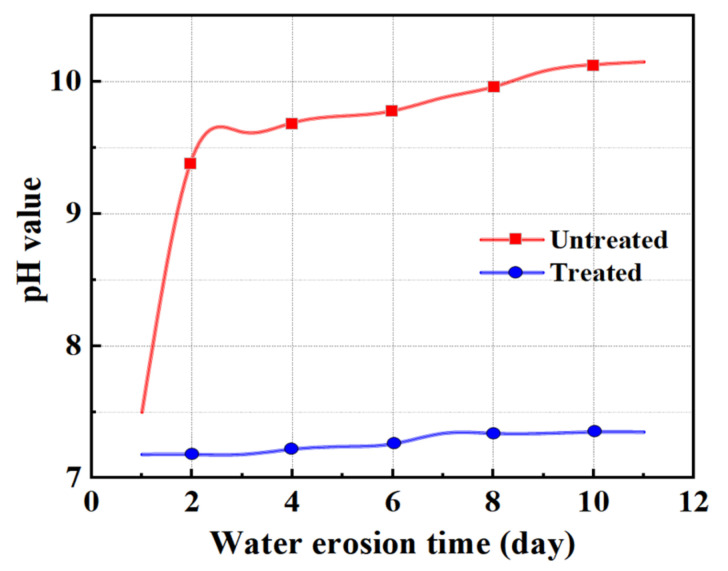
Comparison of pH values during the steel slag water erosion process.

**Figure 8 materials-15-06642-f008:**
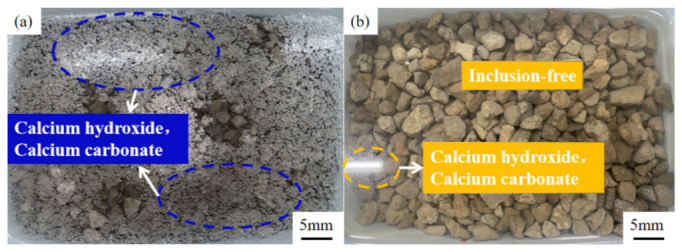
Water erosion process of untreated steel slag: (**a**) untreated; (**b**) treated.

**Figure 9 materials-15-06642-f009:**
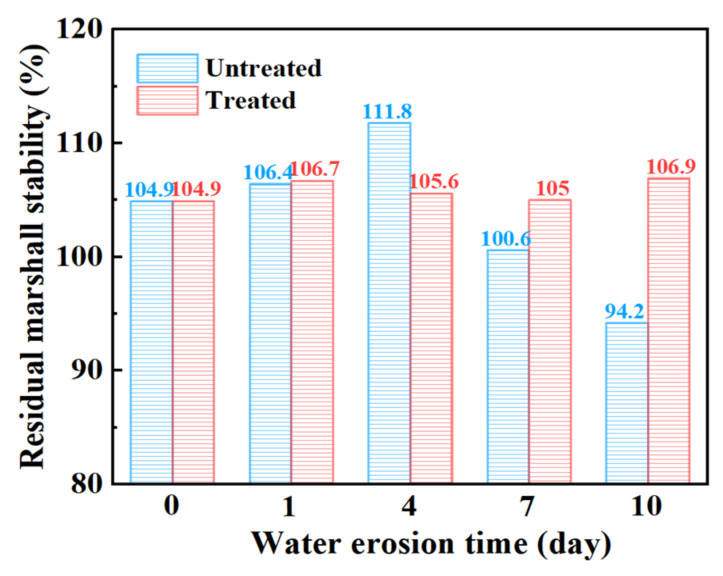
Marshall residual stability of steel slag asphalt mixture with different water erosion times.

**Figure 10 materials-15-06642-f010:**
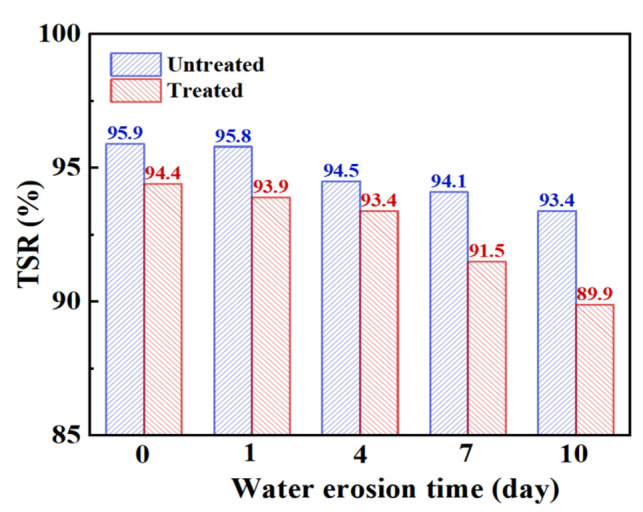
Freeze–thaw splitting strength ratio of steel slag asphalt mixture with different water erosion times.

**Figure 11 materials-15-06642-f011:**
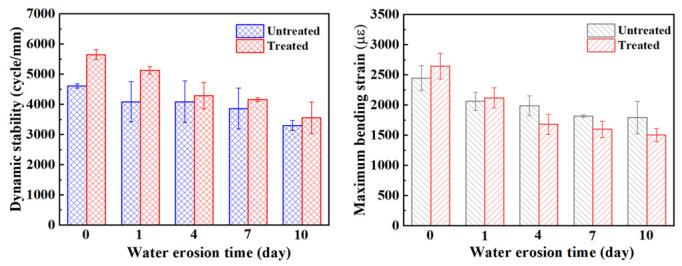
Dynamic stability and maximum bending strain of steel slag asphalt mixture under different water erosion times.

**Figure 12 materials-15-06642-f012:**
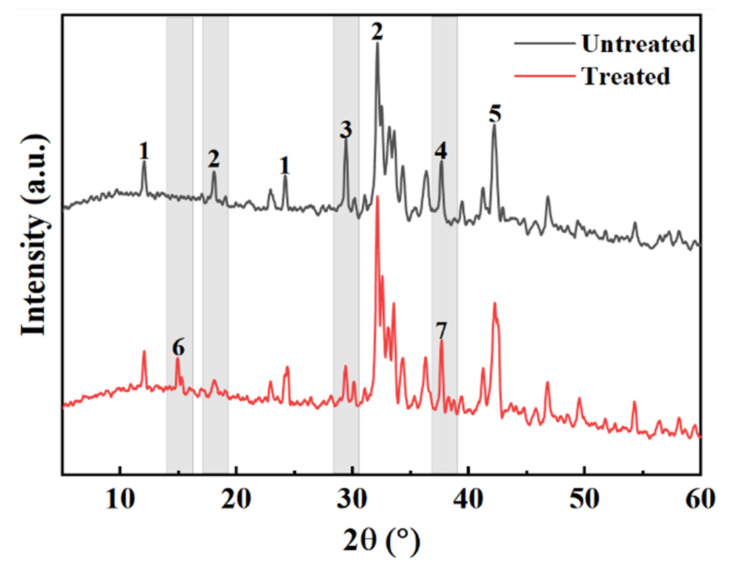
Comparison of mineral composition of steel slag (1—Ca_2_Fe_2_O_5_; 2—Ca_2_SiO_4_; 3—Ca_3_SiO_4_; 4—CaO; 5—RO; 6—CaC_2_O_4_; 7—Ca(OH)_2_).

**Figure 13 materials-15-06642-f013:**
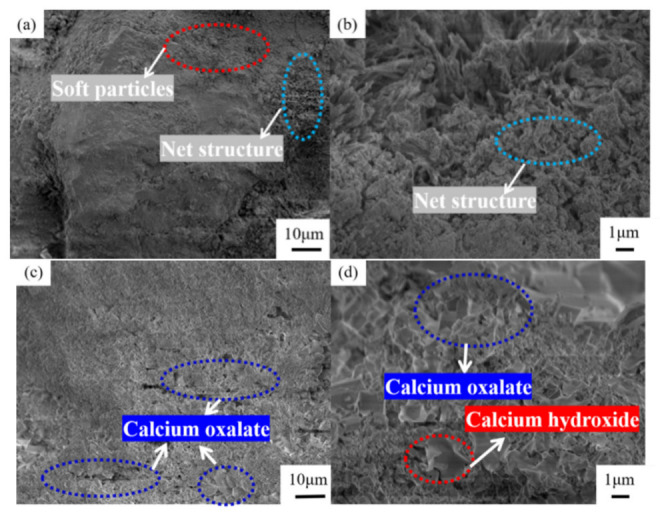
Microstructure of steel slag surface: (**a**,**b**) untreated, (**c**,**d**) treated.

**Figure 14 materials-15-06642-f014:**
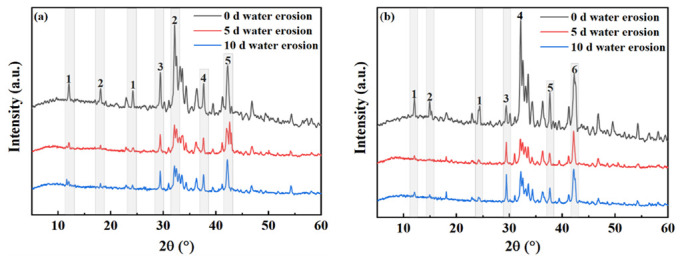
Mineral composition changes of untreated and treated steel slag under different water erosion times. (**a**) Untreated (1—Ca_2_Fe_2_O_5_; 2—Ca_2_SiO_4_; 3—Ca_3_SiO_4_; 4—CaO; 5—RO). (**b**) Treated (1—Ca_2_Fe_2_O_5_; 2—Ca_2_C_2_O_4_; 3—Ca_3_SiO_4_; 4—Ca_2_SiO_4_; 5—Ca(OH)_2_; 6—RO).

**Figure 15 materials-15-06642-f015:**
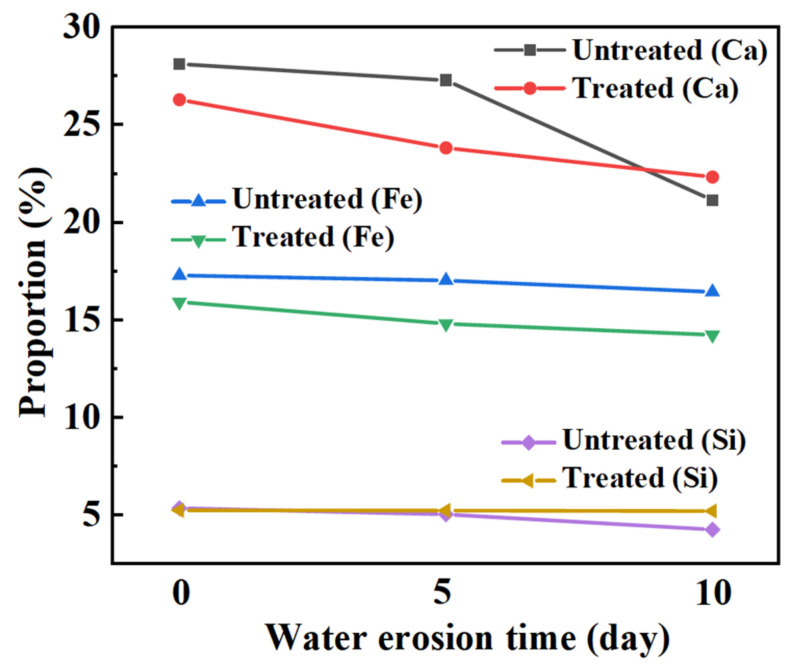
Influence of water erosion on main chemical elements in steel slag.

**Figure 16 materials-15-06642-f016:**
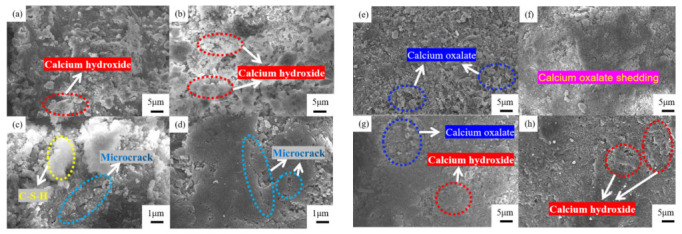
Changes in surface micro-morphology of steel slag under different water erosion times: untreated after (**a**) 1 day, (**b**) 4 days, (**c**) 7 days and (**d**) 10 days; and treated after (**e**) 1 day, (**f**) 4 days, (**g**) 7 days and (**h**) 10 days.

**Table 1 materials-15-06642-t001:** Chemical composition of steel slag (Wt.%).

	CaO	Fe_2_O_3_	SiO_2_	MgO	Al_2_O_3_	P_2_O_5_	MnO
**BOF Steel Slag**	43.49	28.36	12.99	6.75	2.33	2.17	1.34
**SSD [22]**	37.57	27.75	16.35	6.95	3.21	2.69	3.56
**EOF Steel Slag [23]**	35.28	26.91	16.69	9.27	6.2	1.43	1.88

**Table 2 materials-15-06642-t002:** Physical properties of steel slag.

Tested Parameters	Unit	BOF Steel Slag	EOF Steel Slag [23]	Natural Coarse Aggregate [23]	Technical Indicators
**Crush Value**	%	17.8	8	6.55	≤26
**Los Angeles Wear (C)**	%	16.9	1.16	8.27	≤28
**Bibulous Rate**	%	1.91	3	0.8	≤2.0
**Gross Volume Relative Density**	g/cm^3^	3.465	2.86	2.75	—
**Apparent Relative Density**	g/cm^3^	3.586	3.02	2.886	≥2.60
**Adhesiveness**	level	5	-	-	≥5

**Table 3 materials-15-06642-t003:** Physical properties of limestone.

Tested Parameters	Unit	Test Value	Technical Indicators
**Gross Volume Relative Density**	g/cm^3^	2.656	—
**Apparent Relative Density**	g/cm^3^	2.724	≥2.45
**Angularity (Flow Time Method)**	s	38.9	—
**Sand Equivalent**	%	73	≥50

**Table 4 materials-15-06642-t004:** Test results of SBS polymer-modified asphalt.

		Unit	Test Results	Technical Indicators
	**Penetration (25 °C, 5 s, 100 g)**	0.1 mm	53	40~60
	**Penetration Index**	-	0.151	≥0
	**Softening Point (Global Method)**	°C	79.5	≥60
	**Ductility (5 °C)**	cm	29	≥20
	**Flash Point**	°C	294.8	≥230
	**Ignition Point**	°C	304.8	-
	**Solubility**	%	99.76	≥99
	**Elastic Recovery (25 °C)**	%	82	≥75
	**Rotational Viscosity of Brinell (135 °C)**	Pa·s	1.73	≤3
	**Density (15 °C)**	g/cm^3^	1.037	-
	**Relative Density (25 °C)**	-	1.038	-
**Film Heating Test (163 °C, 5 h)**	**Storage Stability Segregation, 48 h Softening Point Difference**	°C	2	≤2.5
	**Quality Change**	%	−0.209	±1.0
	**Penetration Ratio (25 °C)**	%	78	≥65
	**Ductility (5 cm/min, 10 °C)**	cm	21	≥15

**Table 5 materials-15-06642-t005:** Mass passing percentage of graded sieve.

Test Item		Percentage of Mass Passing the Following Sieve (mm)/%								
	16	13.2	9.5	4.75	2.36	1.18	0.6	0.3	0.15	0.075
**AC-13**	100.0	94.7	68.6	41.6	28.4	18.5	15	10.4	7.8	5.1

**Table 6 materials-15-06642-t006:** Marshall test results of steel slag asphalt mixture at optimum asphalt dosage.

Optimum Oilstone Ratio	Gross Volume Relative Density (g/cm^3^)	Void Fraction (%)	Voids in Mineral Aggregate (%)	Effective Asphalt Saturation (%)	Stability (KN)	Flow Value (mm)
**Untreated (5.2%)**	3.0	4.3	17.6	75.6	20.23	2.7
**Treated (4.8%)**	2.96	3.4	14.1	75.7	22.62	3.1

**Table 7 materials-15-06642-t007:** Comparison of f-CaO content (Wt.%).

	Free Calcium	Ca(OH)_2_	f-CaO
**Untreated**	4.852	0.31	4.551
**Treated**	2.848	0.623	2.225

**Table 8 materials-15-06642-t008:** Comparison of chemical composition and basicity of steel slag with limestone (Wt.%).

	CaO	Fe_2_O_3_	SiO_2_	MgO	Al_2_O_3_	MnO	Basicity
**Untreated**	43.49	28.36	12.99	6.75	2.33	1.34	3.35
**Treated**	40.26	25.81	12.54	6.64	2.23	1.19	3.21
**Limestone**	36.27	4.64	13.29	1.62	4.59	1.74	2.73

**Table 9 materials-15-06642-t009:** Test results of stability.

	Soaking Time	0.5 h Marshall Stability (KN)	48 h Marshall Stability (KN)		Soaking Time	0.5 h Marshall Stability (KN)	48 h Marshall Stability (KN)
**Untreated**	0 days	20.11	21.11	**Treated**	0 days	21.81	22.9
	1 day	21.71	23.1		1 day	20.83	22.23
	4 days	19.42	21.72		4 days	19.86	20.98
	7 days	20.94	21.06		7 days	19.68	20.67
	10 days	22.99	21.66		10 days	20.72	22.16

**Table 10 materials-15-06642-t010:** Test results of freeze–thaw splitting strength.

	Water Erosion Time	Freeze–Thaw Cycle Splitting Strength (MPa)	Unfreeze–Thaw Cycle Splitting Strength (Mpa)		Water Erosion Time	Freeze–Thaw Cycle Splitting Strength (Mpa)	Unfreeze–Thaw Cycle Splitting Strength (Mpa)
**Untreated**	0 days	2.11	2.20	**Treated**	0 days	2.37	2.51
	1 d	2.05	2.14		1 day	2.16	2.3
	4 days	1.73	1.83		4 days	1.71	1.83
	7 days	1.77	1.88		7 days	1.62	1.77
	10 days	1.71	1.83		10 days	1.52	1.69

**Table 11 materials-15-06642-t011:** Changes in chemical elements under different water erosion times (Wt.%).

	Ca	Fe	Si	Mg	Al	P	Mn
**Untreated (0 days)**	28.13	17.30	5.38	3.65	1.14	0.83	0.87
**5 days**	27.29	17.04	5.07	3.50	1.11	0.83	0.82
**10 days**	21.16	16.45	4.28	3.42	1.10	0.82	0.81
**Treated (0 days)**	26.30	15.95	5.28	3.63	1.1	0.82	0.79
**5 days**	23.84	14.83	5.26	3.58	1.09	0.8	0.7
**10 days**	22.34	14.26	5.24	3.57	1.08	0.79	0.69

## Data Availability

Not applicable.

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
