# Peer review of "Study on the Performance of Steel Slag and Its Asphalt Mixture with Oxalic Acid and Water Erosion"

_materials, 2022, doi:10.3390/ma15196642_

Round 1

Reviewer 1 Report

The Paper is quite up to the mark but it lacks certain grammatical mistakes like missing some punctuation. The paper is quite structured and below said corrections can be implemented for betterment:

Q1. The more latest literature review can be added in the introduction part including their proper references.

Q2. The Paper can be accepted although some sentences can be shortened and grammatical mistakes can be removed.

Q3. More references can be added to support the result and discussion part in every aspect of influencing factors.

Detailed comments are attached in a word file

Reviewer 2 Report

this paper used self-prepared oxalic acid liquid to treat steel slag in order to prevent volume expansion of steel slag. the steel slag and its asphalt mixture were exposed to water erosion before and after treatment with varying periods of time. the manuscript is well-written and structured and also i has genuine results; however there are some comments should be considered before it is being accepted

Title- please revise the title as the current title doesnt sound well

Abstract- it is very lengthy, please revise it and delete unimportant sentences

Introduction- please cite more recent and relevant studies to show the significance of this study

there are several grammatical errors and typos, Please carefully proof-read spell check to eliminate grammatical errors for  the whole manuscript

for Treatment method, i prefer to see how this processes were conducted by including pics for materials and processes.

Reviewer 3 Report

I have several general concerns and several specific comments.

General Concerns

I will expand on these concerns in the Specific Comments part of my review. 

The manuscript suffers from:

1. Repetitiveness: Ideas are repeated several times within the manuscript in different sections.

2. Vague and at time cryptic assertions and declarations.

3. Lack of organization. The manuscript jumps from Introduction to Materials and Methods. The manuscript need to have different sections for (a) Introduction and Background, (b) Literature Review, (c) Novelty and Objectives of the research, (d) Methodology, (e) Materials, (f) Test Matrix, Number of Specimens Tested for each Test in the Matrix, (g) Test Procedures, (h) Specimen Preparation, (i) Test Results, (j) Analysis of Results, and (k) Conclusions. Organized this way, the reader does not have to search for where each of these important topics are discussed. Otherwise, the reader will be confused and eventually disinterested in reading further the manuscript.

Specific Comments

1. Abstract: There is hardly any introductory remarks in the abstract. The bulk of the Abstract (lines 17 to 35) is presentation of results.

2. There is no discussion anywhere on the novelty of the research and the research gap the research is addressing. The researchers need to demonstrate this important element early on in the manuscript.It is recommended that this is presented after the literature review. Otherwise, the reader will keep asking why is this research performed.

3. As discussed in item 2 above, the literature review is imbedded in the Introduction. 

4. The content from line 61 to line 81 is basically repeated in lines 82 to 97.

5. Discussion from line 88 to line 97 does not belong in the Introduction.

6. Are the tests reported in Tables 1, 2, 3 and 4, performed according to accepted standards? These Standards need to be identified.

7. Lines 118 and 119: The authors need to discuss and present references to these previous tests for the specific treatment processes. The reader may have specific concerns about these processes.

8. Line 131. What do the authors mean by "...the best water content...."? Is this the optimum water content? This is and example of vague and cryptic statement.

9. Lines 133 to the end of the paragraph: The tense suddenly changes from present tense to imperative tense. This change is very unusual. This happens in line 154 as well.

10. Figure 1. What is the surcharge load?

11. Line 188 and the long paragraph that follows. This paragraph is mostly a repetition of materials presented earlier in the manuscript. What is the reason for this repetition? This comment applies also to lines 231 to 237.

12. For all the Figures and Tables presenting the results there is no discussion on the number of repetitions of each test. Are the results averages? If yes, averages of how many repetitions? 

11. Table 8 present equations. Do these equations explain any physical process? Or they are mere exercises in regression.

12. What is the justification for choosing 10 days as the maximum erosion time? Extended erosion time could have given more insight in the long-term properties.

Reviewer 4 Report

The paper deals with the Study on the Influence Mechanism of Oxalic Acid to Performance of Steel Slag and Its Asphalt Mixt. 

According to the reviewer, the paper is not worth publishing at Materials Journal, 

since corrections are needed and then the paper can be accepted for publication in the journal.

While the authors have made considerable research effort, 

the presentation of the paper and the results must be proved. 

Additionally make the following corrections to the manuscript:

Comment 1

Figures 2, 7 and 11 are not visible enough!!!

Major Problem.

Comment 2

Increase the quality of the References (and the number of the reference papers including (primarily) from MDPI journals).

The authors use 0 paper Materials / 0 paper from MDPI journals / 44 papers from journals (References)

Τhe number for papers from MDPI journals 

is considered insufficient (in reviewer's opinion).

Also delete the [J] and [D].

The authors must format the References according to the journal's instructions.

Comment 3

Extended text editing.

No Line numbering.

Round 2

Reviewer 1 Report

NO further comments

Author Response

Thank the reviewers for their hard work.

Reviewer 3 Report

The authors have addressed my review comments satisfactorily.

Author Response

(The authors gave the same response as above.)

Reviewer 4 Report

Comment 1

Extended text editing

Line 49

[6~9]

The authors must replace

[6~9].

Line 116

The specific treatment process is as follows,

The authors must insert in the text the Figure 1.

Figure 4

Untreaeted

Line 532

Followig

Lines 601 - 645

The authors must format the text according to the journal's instructions.

Comment 2

Lines 81 and 600

It is not so good to start a Section in the end of the page without text.

The authors must format.

Lines 194 - 195

It is not so good to start a Section in the end of the page without text.

The authors must format.

Figure: 4, 5, 6, 13, 14

The Figure must be accompanied on the same page as the Figure's title.

Comment 3

Lines 120, 129 and 181 

It is so good to use the word "we".

The authors must rephrase.

Comment 4

Lines 150 - 152

The authors must format (full alignment).

Comment 5

The authors must give more details for using experimental equipment (type, model).
